# Impacts of mild COVID-19 on elevated use of primary and specialist health care services: A nationwide register study from Norway

Katrine Damgaard Skyrud[1]*, Kjersti Helene Hernæs[1], Kjetil Elias Telle[1], Karin Magnusson[1,2]

1 Norwegian Institute of Public Health, Cluster for Health Services Research, Oslo, Norway, 2 Faculty of Medicine, Department of Clinical Sciences Lund, Orthopaedics, Clinical Epidemiology Unit, Lund University, Lund, Sweden

* katrinedamgaard.skyrud@fhi.no

**Data Availability Statement:** Data cannot be shared publicly because individual-level data of

## Abstract

### Aim

To explore the temporal impact of mild COVID-19 on need for primary and specialist health care services.

### Methods

In all adults (≥20 years) tested for SARS-CoV-2 in Norway March 1st 2020 to February 1st 2021 (N = 1 401 922), we contrasted the monthly all-cause health care use before and up to 6 months after the test (% relative difference), for patients with a positive test for SARS-CoV-2 (non-hospitalization, i.e. mild COVID-19) and patients with a negative test (no COVID-19).

### Results

We found a substantial short-term elevation in primary care use in all age groups, with men generally having a higher relative increase (men 20–44 years: 522%, 95%CI = 509–535, 45–69 years: 439%, 95%CI = 426–452, ≥70 years: 199%, 95%CI = 180–218) than women (20–44 years: 342, 95%CI = 334–350, 45–69 years = 375, 95%CI = 365–385, ≥70 years: 156%, 95%CI = 141–171) at 1 month following positive test. At 2 months, this sex difference was less pronounced, with a (20–44 years: 21%, 95%CI = 13–29, 45–69 years = 38%, 95% CI = 30–46, ≥70 years: 15%, 95%CI = 3–28) increase in primary care use for men, and a (20–44 years: 30%, 95%CI = 24–36, 45–69 years = 57%, 95%CI = 50–64, ≥70 years: 14%, 95%CI = 4–24) increase for women. At 3 months after test, only women aged 45–70 years still had an increased primary care use (14%, 95%CI = 7–20). The increase was due to respiratory- and general/unspecified conditions. We observed no long-term (4–6 months) elevation in primary care use, and no elevation in specialist care use.

patients included in this manuscript after de-identification are considered sensitive.

**Funding:** The study was funded by the Norwegian Institute of Public Health. No external funding was received. This does not alter our adherence to PLOS ONE policies on sharing data and materials.

**Competing interests:** The study was funded by the Norwegian Institute of Public Health. No external funding was received. This does not alter our adherence to PLOS ONE policies on sharing data and materials.

## Conclusion

Mild COVID-19 gives an elevated need for primary care that vanishes 2–3 months after positive test. Middle-aged women had the most prolonged increased primary care use.

## Introduction

Patients being hospitalized with COVID-19 may suffer multiorgan failure or other physical problems, as well as psychological dysfunctions [1–3]. In addition to an imminent risk of severe health impairments and death within weeks after infection, the disease is believed to worsen health for months or years after infection, with an expected need for rehabilitation services [4–10]. However, we have poor knowledge of how COVID-19 affects the vast majority of surviving patients who were not hospitalized, and whether they will become an excessive burden on primary and specialist health care services for a long time.

Though post-acute or long covid is not clearly defined and still poorly documented, there have been reports of patients with symptoms for at least 6 months after initial infection, also among those with mild initial infection. The most commonly reported symptoms after mild disease are cough, low grade fever, fatigue, shortness of breath, chest pain, headaches, neuro-cognitive difficulties, muscle pains and weakness, gastrointestinal upset, skin rashes, metabolic disruption, thromboembolic conditions, and mental health conditions [2, 3, 11]. A recent study from Denmark also reported increased risk of complications in the overall six-months period following initial mild disease (not hospitalized) [12].

In summary, a wide variety of post-covid symptoms are listed for different severity grades in a limited number of study populations, calling for more knowledge about the temporal pattern of long-term morbidities. When high-risk groups with an expected high burden on health care to an increasing extent are vaccinated, learning more about the impact of *mild* COVID-19 on the different levels in health care will be increasingly important. Such knowledge of the consequences of undergoing mild COVID-19 is important in deciding to terminate or continue strict lockdown and disease control measures.

To enable the protection of specific age and sex groups in such future disease control measures, detailed knowledge of the duration and burden on health care is required. Thus, we aimed to explore the temporal patterns of elevated need for primary and specialist health care services for up to 6 months after testing positive for SARS-CoV-2, in particular for the vast majority of surviving patients who were not hospitalized with COVID-19, stratified by age and sex.

## Methods

### Design & data sources

Applying an observational pre-post design with a comparison group, we utilized population-wide longitudinal registry data from Norway to show health care utilization month by month before and after PCR-test for COVID-19. The BeredtC19-register is an emergency preparedness register aiming to provide rapid knowledge about the pandemic, including impacts of measures to limit the spread of the virus on health and utilization of health care services [13]. BeredtC19 compiles daily updated individual-level data from several registers, including the Norwegian Surveillance System for Communicable Diseases (MSIS) (all testing for COVID-19), the Norwegian Patient Register (NPR) (all electronic patient records from all hospitals in

Norway), and the Norway Control and Payment of Health Reimbursement (KUHR) Database (all consultations with all general practitioners and emergency primary health care) as well as the National Population Register (age, sex, country of birth, date of death). Thus, the register includes all polymerase chain reaction (PCR) tests for COVID-19 in Norway with date of testing and test result, reported from all laboratories in Norway to MSIS and all electronic patient records from primary care as well as outpatient and inpatient specialist care. Anti-gen tests were rarely used before February 2021, which is when our study period ends, and all positive anti-gen tests were followed up with a PCR test and thus included in our analyses. The establishment of an emergency preparedness register forms part of the legally mandated responsibilities of The Norwegian Institute of Public Health (NIPH) during epidemics. Institutional board review was conducted, and The Ethics Committee of South-East Norway confirmed (June 4th 2020, #153204) that external ethical board review was not required.

## Population

Our population included every adult ($\geq$20 years) resident of Norway on January $1^{st}$ 2020—as well as everyone born in 2020—who had been tested for SARS-CoV-2 by a PCR-test from March $1^{st}$ 2020 to February $1^{st}$ 2021 (non-residents like tourist etc. excluded). With outcome data that were updated daily with new health records, from January $1^{st}$ 2020 through May $17^{th}$ 2021, we followed all persons for at least 2 months before and at least 3 months after the test date. A large proportion could be followed for 3 months before and through 6 months after (otherwise censored).

## COVID-19

We studied all adults ($\geq$20 years) with a PCR-tests for SARS-CoV-2 in Norway, divided into two mutually exclusive groups:

1. No COVID-19, comprising all individuals with a negative PCR test, who had no routine visits in specialist care in the same week or in the $1^{st}$ or $2^{nd}$ week following the test (comparison group). For individuals with multiple negative tests and no positive test, we chose the first test date.

2. Mild COVID-19, comprising all individuals with a positive PCR test, again, who had no routine visits in specialist care in the same week or in the $1^{st}$ or $2^{nd}$ week following the test. The positive test may or may not be preceded by one or more negative tests. For the very few individuals with several positive tests, we used the first.

   In an own sub-group analysis, we also studied the group having severe COVID-19, which comprised all individuals with a positive PCR test who did visit specialist care during the $0^{th}$ to $2^{nd}$ week following the test week.

## Outcomes

We studied all-cause utilization of primary and specialist care in the 1–24 weeks (i.e. 1–6 months) after the test week. For the different health care levels, our all-cause outcomes were two categorical variables set to one if the person had 1) visited *primary care* (i.e. general practitioners or emergency wards) at least once during a week, or 2) visited specialist care (i.e. hospital-based *outpatient or inpatient specialist care)* at least once during a week. If health care use at one of the health care levels was increased following mild COVID-19, we also studied potential causes for the increase, using the International Classification of Primary Care (ICPC-2), as described in S1 Table (causes: digestive, circulatory, respiratory, endocrine/metabolic/

nutritional, genitourinary, eye/ear, musculoskeletal, mental, skin, blood and general/unspecified conditions).

## Statistical analyses

We first studied the percent using health care services at least once per week from 3 months prior to test week, to 6 months after the test week for persons with mild and no COVID-19, overall and by groups of age and sex. Thus, we calculated the percent using health services per calendar week, and presented averages over the 3 months before test week, and over post-test periods 1–4 weeks, 5–8 weeks, 9–12 weeks and 13–24 weeks. We also plotted the percentages by periods of four weeks following the test week (these were adjusted for potential confounders as described below).

Second, to estimate how much larger or smaller the use of health care services was for those with mild COVID-19 compared with those with no COVID-19, we used a generalized difference-in-differences (DiD) approach. DiD analysis evaluates the effect of an event by comparing the change in outcome for the affected group before and after the event, to the change over the same time span in a group not affected by the event [14–16]. In this study, we compared the rate of health care use in the months before and after the PCR test for those with mild COVID-19 (difference 1), to the difference in the rate of health care use in the months before and after the PCR test for those with no COVID-19 (difference 2). The DiD estimate is the difference between these two differences, estimated using linear probability models with robust standard errors and presented as a difference in percentage points. Statistically, one uses an interaction term (between pre-post PCR-tests and COVID-19 group category) to derive the DiD estimate. By including calendar month fixed effects, this approach accounts for background trends like seasonal variations in health care use [14]. The DiD estimate can be interpreted as the change in health care use that is related to mild COVID-19, beyond any background calendar month trends. If there is no relationship between mild COVID-19 and subsequent health care use, the DiD estimate would be zero.

We generalized this traditional DiD method by extending from one to four post-test periods: 1–4 weeks, 5–8 weeks, 9–12 weeks and 13–24 weeks, comparing to one pre-period (the 3 months before test week) and also including a separate parameter for the test week. The generalization was implemented by including categorical variables for each of these extra periods and accompanying interaction terms. In addition to the presentation of results as absolute differences in percentage points, we also presented relative differences (i.e. in percent) by dividing the absolute estimate (and corresponding lower and upper confidence interval bounds) for each of the post periods by the health care use rate of the comparison group in the pre period (and multiplying by 100).

DiD models are used in two data situations, one where *different* individuals are studied before and after the event, and another, which is our case, where the *same* individuals are followed from before to after the event [15]. While adjusting for individual characteristics that are constant over time can be important in the first type of data situation to account for changes in composition, it is less likely to affect our DiD estimates where the same individuals are followed over time. However, composition might also matter in our situation due to the censoring and adjusting may improve precision [15]. We therefore adjusted for the following individual characteristics: Age (groups 20–44, 45–69, 70 or older) (if relevant), sex (women/men) (if relevant), comorbidities (categories 0, 1, 2 or 3 or more comorbidities) based on risk conditions for COVID-19 defined by an expert panel [17], birth country (Norway/abroad) and calendar month (12 categories).

Most models were run separately by sex and age groups (20–44 years, 45–69 years and 70 or older), as well as for each element in the two groups of outcomes: all-cause health care use in primary and specialist care, and a number of cause-specific health care use as described in S1 Table, i.e. following the broad diagnosis chapters of ICD-10 and ICPC-2. Some patients may contact the primary care physician to certify sickness absenteeism from work, and to explore the robustness of our results, we also undertook an analysis where we excluded consultations that resulted in the physician providing a sick note. Persons who died were censored from the date of death in all analyses. For completeness, we also studied persons with severe COVID-19 applying the DiD-models as described above, and estimated the proportion of deaths within 3 and 6 months after test week for those with no, mild and severe COVID-19 (95% CIs were calculated based on Wilson). All analyses were run in STATA SE v.16.

## Results

We studied every person 20 years or older in Norway (N = 1 401 922) who had been tested for SARS-CoV-2 in Norway from March 1st 2020 to February 1st 2021. In total, 42 313 (3.0%) patients tested positive and had mild COVID-19 (i.e. were not hospitalized), whereas 2 857 (0.2%) patients tested positive and had severe disease (i.e. were hospitalized).

Table 1 shows that the persons testing positive and negative for SARS-CoV-2 were similar in terms of age and comorbidities. However, persons who tested positive and had mild disease were more often born abroad compared to persons who tested negative (Table 1).

Among those experiencing mild COVID-19, 0.74% (CI: 0.67–0.83) died within 3 months, whereas 0.30% (0.29–0.31) died within 3 months after testing negative. The corresponding numbers for death within 6 months were 0.85% (0.77–0.94) for mild COVID-19 and 0.49% (0.48–0.51) for no COVID-19. Among persons with severe disease COVID-19, 9.73% (8.69–10.87) died within 3 months and 10.92% (9.83–12.11) died within 6 months after tested.

### Group-wise change in rates of health care use following test week

About 6% of the patients with mild COVID-19 used *primary care* at least once per week in the 12–1 weeks before testing positive for SARS-CoV-2, and this rose to 27% in the 1–4 weeks after the test, before gradually decreasing at 5–8 weeks (8.5%) and 9–12 weeks after testing (6.9%) (Table 2). By 13–24 weeks after the test week, the percent using primary care had dropped back to 6% (Table 2). Persons with no COVID-19 had no such increase, with 5%

**Table 1. Descriptive characteristics.**

| | Age 20–44 | | Age 45–69 | | Age 70 or older | |
|---|---|---|---|---|---|---|
| | Mild COVID-19 | No COVID-19 | Mild COVID-19 | No COVID-19 | Mild COVID-19 | No COVID-19 |
| Women | | | | | | |
| Population, n | 11630 | 408144 | 6733 | 254804 | 1511 | 53638 |
| Age, mean (SD) | 31(7.25) | 32(7.08) | 54(6.46) | 55(6.73) | 81(8.12) | 79(7.60) |
| Born abroad, % | 40.6 | 18.8 | 35.0 | 14.0 | 9.5 | 5.3 |
| ≥2 comorbidities, % | 0.4 | 0.5 | 2.9 | 2.7 | 12.7 | 13.2 |
| Men | | | | | | |
| Population, n | 13634 | 380575 | 7718 | 219187 | 1087 | 40404 |
| Age, mean (SD) | 31(7.09) | 32(7.02) | 54(6.54) | 55(6.73) | 77(6.53) | 77(6.47) |
| Born abroad, % | 41.1 | 20.7 | 36.9 | 18.2 | 12.9 | 5.9 |
| ≥2 comorbidities, % | 0.3 | 0.3 | 4.2 | 3.5 | 18.6 | 19.2 |

**Table 2. Percent of tested persons who used health care services per week in given time periods before and after PCR test for SARS-CoV-2, by use of primary and specialist (outpatient or inpatient) care during, separately for those with no or mild COVID-19.**

|  | 12–1 weeks pre test | Test week | 1–4 weeks post test | 5–8 weeks post test | 9–12 weeks post test | 13–24 weeks post test |
|---|---|---|---|---|---|---|
| Primary care |  |  |  |  |  |  |
| No COVID-19 | 5.2 | 24.0 | 7.4 | 6.3 | 6.3 | 6.2 |
| Mild COVID-19 | 6.0 | 51.0 | 27.4 | 8.5 | 6.9 | 5.8 |
| Specialist care |  |  |  |  |  |  |
| No COVID-19 | 1.8 | 0.0 | 1.0 | 2.1 | 2.3 | 2.8 |
| Mild COVID-19 | 2.0 | 0.0 | 1.0 | 2.2 | 2.2 | 2.4 |

using services per week in the 12–1 weeks before test, 7% in the 1–4 weeks after the test and 6% in the 13–24 weeks after the test (Table 2).

No increased rates of *specialist care* were observed among the persons testing positive for SARS-CoV-2, nor among the persons testing negative (Table 2). Since we excluded persons who were hospitalized (inpatient or outpatient) in the test week or in the 1 or 2 weeks following the test week, the zero specialist consultations observed in the test week (Table 2) occur by deliberate construction of the study population.

Similar patterns, i.e. with a steep rise in primary, but not specialist care use following a positive test for SARS-CoV-2, were confirmed in sex-and age-specific plots (20–44, 45–69 and 70 + years) that were adjusted for birth country, comorbidities and calendar month (Fig 1). In general, the level and trend in primary and specialist care services use before infection were very similar for those with mild and no COVID-19, and about 2 months after the test those with mild COVID-19 had the same or even lower health care use than those with no COVID-19 (see Fig 1).

### Impact of COVID-19 on primary and specialist care use for women

When comparing the groupwise changes over time with each other, we observed a large elevation in primary care at 1–4 weeks following mild COVID-19 (342% relative increase for women aged 20–44, 375% for women aged 45–69 and 156% for the eldest women) (Table 3). The percent relative difference was reduced in the 5–8 weeks after the test to 30% (age 20–44), 57% (age 45–69) and to 14% for the eldest women (Table 3). In the 9–12 weeks after the test week, only the women aged 45–69 had elevated utilization of primary care (Table 3). We observed no increased use of specialist care following mild COVID-19, for none of the studied post-covid time periods (Table 3). As expected, women with severe COVID-19 had a larger and more prolonged increase in primary care use, and also an increased specialist care use than was observed for women with mild COVID-19 (i.e. when compared to the same comparison group–testing negative and not being hospitalized) (S3 Table).

### Impact of COVID-19 on primary and specialist care use for men

We observed a similar pattern for men as for women, with a sharp increase in primary care use at 1–4 weeks following positive test and having mild disease (Table 4). At 5–8 weeks post-test, the increase had declined to 21% (men 20–44 years), 38% (men 45–69 years) and 15% (men 70 years and older) (Table 4). We observed no increased primary care use at 9–12 weeks nor at 13–24 weeks following positive test and having mild disease (Table 4). Also, we observed no increased specialist care following positive test and mild disease (Table 4). However, as expected, men with severe COVID-19 had a larger and more prolonged increase in primary care use, and also an increased specialist care use than was observed for men with mild

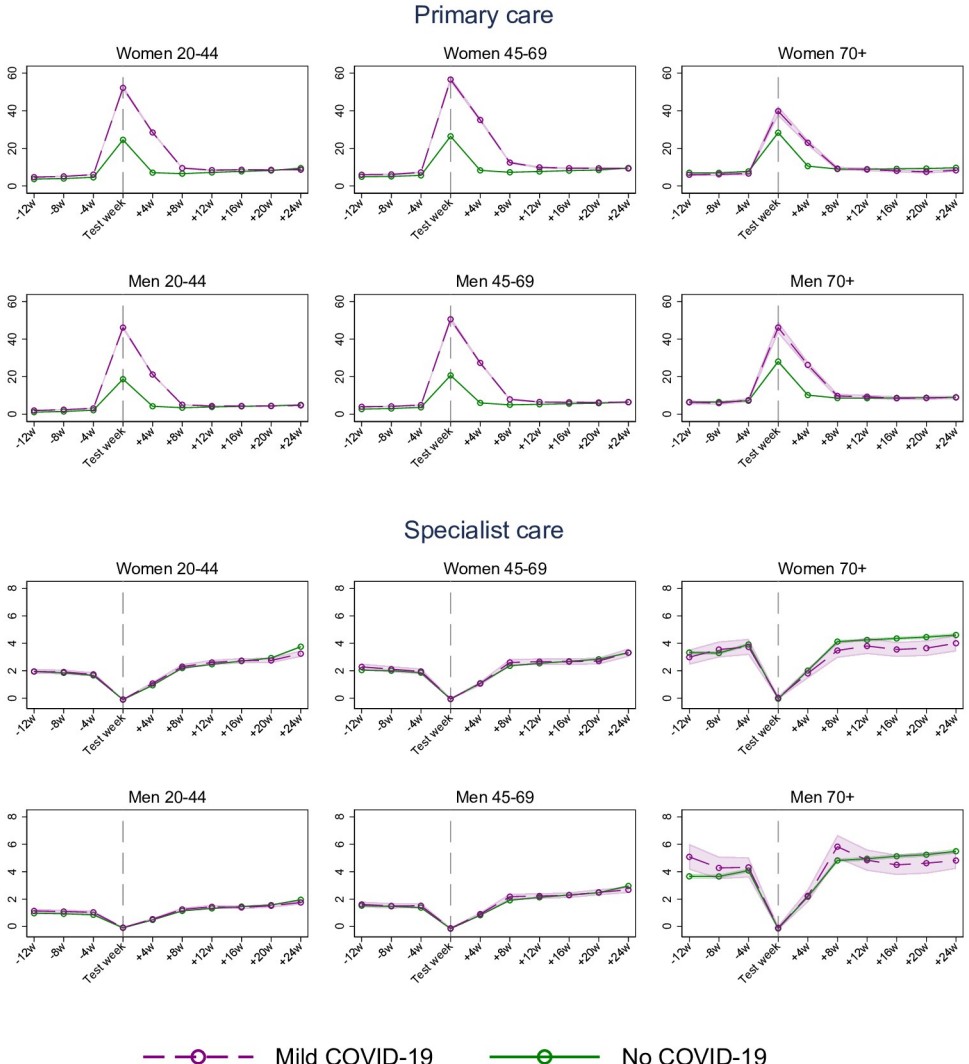

**Fig 1. Estimated percent (95% CI) of women and men using primary and specialist (inpatient and outpatient) care in a week, from 3 months before to 6 months after week of PCR test for SARS-CoV-2, for persons with mild COVID-19 and no COVID-19, by age groups.** Estimates adjusted for comorbidities, birth country and calendar month. The dip for specialist care around the test week is a mechanical result of the exclusion of persons who were hospitalized with COVID-19 in the test week and the two following weeks.

COVID-19 (i.e. when compared to the same comparison group–testing negative and not being hospitalized) (S3 Table). Thus, the patterns of increased health care use following severe COVID-19 were similar for men and women (S3 and S4 Tables).

## Age- and sex-independent causes for increased primary care use

Fig 2 shows a similar sudden increase and steep decline in primary care use for visits that were due to respiratory conditions, i.e. corresponding to the increase and decline seen for all-cause primary care use (Fig 1). Also, a similar pattern can be observed for general and unspecified conditions (typically covering diagnostic codes for fatigue, cough, myalgia etc.). Thus, altogether, our results suggest that respiratory conditions and general and unspecified conditions may be the main causes for the increased primary care use following mild COVID-19 (Fig 2).

**Table 3. Difference-in-differences (DiD) estimates of impacts of COVID-19 on health care use from 12–1 weeks before PCR test for SARS-CoV-2 to 1–4, 5–8, and 9–12 weeks after as well as 13–24 weeks after PCR test.**

| | Women 20–44 | | Women 45–69 | | Women 70 + | |
|---|---|---|---|---|---|---|
| | B (95% CI) | % relative diff. (95% CI) | B (95% CI) | % relative diff. (95% CI) | B (95% CI) | % relative diff. (95% CI) |
| Primary care | | | | | | |
| 1–4 weeks | 20.17 (19.69,20.65) | 342 (334,350) | 25.58 (24.90,26.26) | 375 (365,385) | 13.41 (12.130,14.68) | 156 (141,171) |
| 5–8 weeks | 1.75 (1.41, 2.10) | 30 (24,36) | 3.90 (3.40, 4.39) | 57 (50,64) | 1.21 (0.362, 2.06) | 14 (4,24) |
| 9–12 weeks | 0.009 (-0.30, 0.32) | 0 (-5,5) | 0.94 (0.51, 1.36) | 14 (7,20) | 0.80 (-0.055, 1.65) | 9 (-1,19) |
| 13–24 weeks | -1.36 (-1.60,-1.12) | -23 (-27,-19) | -0.90 (-1.22,-0.59) | -13 (-18,-9) | -0.39 (-1.022, 0.24) | -5 (-12,3) |
| Specialist care | | | | | | |
| 1–4 weeks | 0.09 (-0.05, 0.22) | 5 (-2,12) | -0.18 (-0.35,-0.002) | -9 (-17,0) | -0.12 (-0.59, 0.348) | -3 (-16,9) |
| 5–8 weeks | 0.06 (-0.11, 0.23) | 3 (-6,12) | 0.097 (-0.14, 0.337) | 5 (-7,16) | -0.57 (-1.15, 0.016) | -15 (-30,0) |
| 9–12 weeks | 0.08 (-0.10, 0.27) | 4 (-5,14) | -0.029 (-0.27, 0.216) | -1 (-13,10) | -0.36 (-0.99, 0.259) | -10 (-26,7) |
| 13–24 weeks | -0.35 (-0.52,-0.18) | -18 (-27,-9) | -0.18 (-0.40, 0.048) | -9 (-19,2) | -0.60 (-1.11,-0.095) | -16 (-29,-3) |

The DiD estimates captures the change in health care use from 12–1 weeks before PCR test to 1–4, 5–8, and 9–12 weeks after as well as 13–24 weeks after PCR test for *female* patients with *mild* COVID-19 compared with the change over the same period for women with no COVID-19.

## Discussion

### Principal findings

In this population-wide study of 1 401 922 persons comprising everyone tested for SARS-CoV-2 in Norway by February 1st 2021, we find that mild COVID-19 gives an elevated need for primary care that vanishes 2–3 months after positive test. Middle-aged women had the most prolonged increased primary care use, which persisted for up to 3 months. The increased primary care use was due to respiratory conditions and general and unspecified conditions, and we observed no increased specialist care for patients who were mildly affected by COVID-19.

To our knowledge, we are the first to document such a time-restricted elevated health care use following a mild disease course of COVID-19. With our registry-based prospective design,

**Table 4. Difference-in-differences (DiD) estimates of impacts of COVID-19 on health care use from 12–1 weeks before PCR test for SARS-CoV-2 to 1–4, 5–8, and 9–12 weeks after as well as 13–24 weeks after PCR test.**

| | Men 20–44 | | Men-45-69 | | Men 70 + | |
|---|---|---|---|---|---|---|
| | B (95% CI) | % relative diff. | B (95% CI) | % relative diff. | B (95% CI) | % relative diff. |
| Primary care | | | | | | |
| 1–4 weeks | 15.96 (15.57,16.35) | 522 (509,535) | 20.09 (19.51,20.67) | 439 (426,452) | 16.16 (14.61,17.70) | 199 (180,218) |
| 5–8 weeks | 0.65 (0.41, 0.88) | 21 (13,29) | 1.74 (1.36, 2.12) | 38 (30,46) | 1.26 (0.22, 2.30) | 15 (3,28) |
| 9–12 weeks | -0.42 (-0.63,-0.20) | -14 (-20,-7) | 0.04 (-0.29, 0.37) | 1 (-6,8) | 0.90 (-0.13, 1.92) | 11 (-2,24) |
| 13–24 weeks | -1.05 (-1.21,-0.88) | -34 (-40,-29) | -0.98 (-1.24,-0.71) | -21 (-27,-16) | 0.006 (-0.75, 0.76) | 0 (-9,9) |
| Specialist care | | | | | | |
| 1–4 weeks | -0.12 (-0.22,-0.02) | -11 (-20,-2) | -0.004 (-0.16, 0.15) | 0 (-9,9) | -0.69 (-1.40, 0.01) | -16 (-33,0) |
| 5–8 weeks | -0.07 (-0.19, 0.06) | -6 (-18,5) | 0.18 (-0.04, 0.39) | 10 (-2,23) | 0.26 (-0.63, 1.14) | 6 (-15,27) |
| 9–12 weeks | -0.06 (-0.19, 0.08) | -6 (-18,7) | 0.007 (-0.21, 0.22) | 0 (-12,13) | -0.85 (-1.72, 0.008) | -20 (-41,0) |
| 13–24 weeks | -0.31 (-0.42,-0.19) | -29 (-39,-18) | -0.24 (-0.42,-0.05) | -14 (-25,-3) | -1.40 (-2.12,-0.68) | -33 (-50,-16) |

The DiD estimates captures the change in health care use from 12–1 weeks before PCR test to 1–4, 5–8, and 9–12 weeks after as well as 13–24 weeks after PCR test for *male* patients with *mild* COVID-19 compared with the change over the same period for men with no COVID-19.

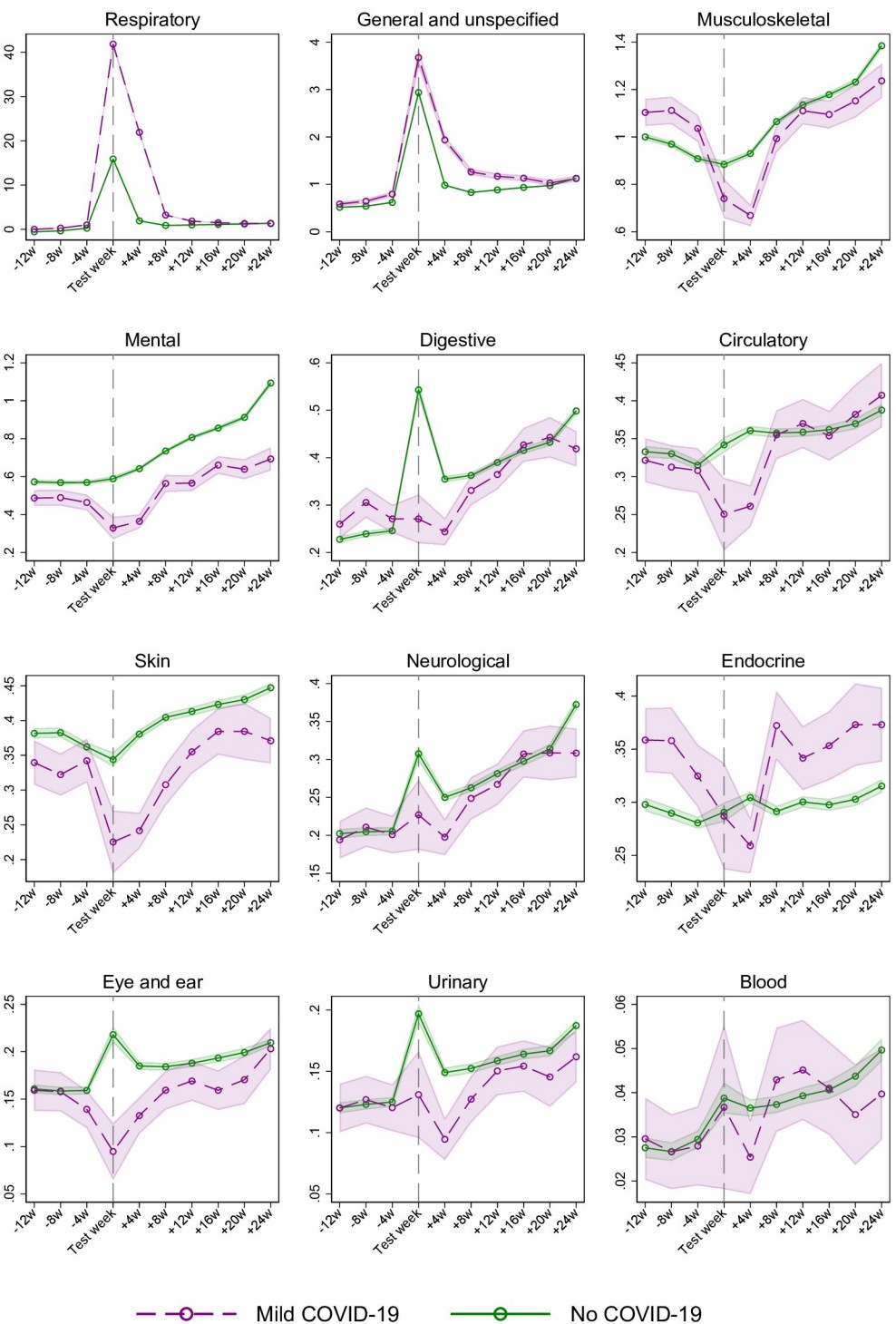

**Fig 2. Estimated percent (95% CI) of women and men using primary care in a week, from 3 months before to 6 months after week of PCR test for SARS-CoV-2, for persons with mild COVID-19 and no COVID-19, by cause-specific diagnosis groups.** Estimates adjusted for age, sex, comorbidities, birth country and calendar month. The cause-specific diagnosis conditions are ordered by decreasing percentage of primary care.

we have also for the first time shown *how* COVID-19 patients are affected and *when* they use *which type* of health care, i.e. we could study both primary- and specialist care use for any condition affecting internal-, external-, sensory- and other organs. These findings are important to report, considering the current stage of the pandemic in many countries. When persons at risk are vaccinated, more knowledge is needed regarding the short- and long-term burden of *mild* disease.

## Comparison with related studies

While the long-term deterioration in health for patients who were hospitalized with COVID-19 has been documented in several studies [3, 5, 9, 10], and also observed in a sub-analysis in the current study (S3 and S4 Tables), evidence on health care use following mild COVID-19 is limited [12]. The long COVID or post-acute COVID syndrome has been described as a multi-system disease (based on patient-reported data), sometimes occurring after a relatively mild acute illness, i.e. extending beyond 3 months after the onset of the first symptoms [2]. Typical symptoms included in this syndrome are fatigue, cough and musculoskeletal pain. We indeed find a minor (4%) increase in visits due to "general and unspecified conditions" in primary care in the first 3 months after positive test, which could typically include fatigue, cough and musculoskeletal pain and other complaints.

These findings add to the recently published Danish register study showing a small increased risk of post-viral complications within 6 months after having undergone mild COVID-19 [12]. Here, we additionally show a limited duration of such post-viral complications following mild disease, i.e. that elevated health care use vanishes 2–3 months after a positive test. We could also demonstrate some minor sex differences in health care use following mild disease, the most remarkable being the larger increase in men than in women during the first month, and the more prolonged increase for middle-aged women (age 45–69), lasting up to 3 months, compared to 2 months for middle-aged men. The causes for these sex differences are unclear, yet may be related to differences in the severity grade of complaints experienced by men vs. women, differences in how the disease affects men and women, but also by differences in health-care seeking behavior, which is known to greatly vary with age and sex [18, 19]. Our study revealed a higher infection rate for persons born outside Norway (Table 1), and our findings are supported by other European studies [20, 21]. Labberton et al. show that COVID-19 lead to higher rates of infection and hospitalization in Norway's immigrant population [22]. Thus, although country of origin remains a relevant factor when studying health-care use, the subject deserves a more thorough investigation, which is beyond the scope of this article. Finally, our study may shed new light on recent reports regarding the content of health care visits following mild COVID-19, as we find that the increase in primary care use could be explained by a simultaneous increase in visits due to respiratory conditions and general/unspecified conditions (Fig 2). Given results in previous studies [2, 4], the increase in general and unspecified conditions in primary care deserves attention in future research.

## Interpretation

In addition to including everyone tested for SARS-CoV-2 in an entire nation with high testing rates, our rich data allowed us to follow all individuals, week-by-week, from 3 months before to 6 months after the PCR test, with no attrition (except death, for which we censored by date). The longitudinal data allowed us to use the difference-in-differences (DiD) design, so that we could adjust health care use after infection with health care use before infection for the same individuals. We could also use individuals testing negative for SARS-CoV-2 as a comparison group to adjust for monthly variation in health care use during the pandemic. The attractive

features of DiD becomes particularly evident for our findings about use of health care for mental conditions (Fig 2). Whereas mental health visits more than doubled from 3 months before to 6 months after the PCR test, persons with no COVID-19 suffered a similar or even more pronounced increase in mental conditions over the same period (Fig 2). This pattern suggests that it is the pandemic and its countermeasures–not actually undergoing COVID-19 –that induces mental distress. The DiD model accounts for such period effects [14–16], and thus provides no estimates of positive impacts of COVID-19 on mental health care visits [23]. The example with mental distress underlines the need for always including a comparison group when studying e.g. depressive mood or anxiety for patients who have undergone COVID-19 [3].

Of further importance to the interpretation of our findings are the high rate of primary care use during the test week. These peaks are expected, as we also include digital general practitioner (GP) consultations (phone, video-or e-mail consultations). About 50% of all consultations in the test week were digital, compared to 30% thereafter, and were more frequent in the younger age groups. For example, digital GP consultations are likely to increase when COVID-19 is suspected and patients are quarantined or isolated. In addition, in the spring of 2020 the doctors' payments for conducting digital consultations rose to the same level as traditional consultations, and it was allowed to issue sick notes in digital consultation [24]. We were not able to distinguish between phone, video-or e-mail consultations. Along this line, parts of the increase in health care use during our post-test periods may be explained by a routine follow-up visit that is recommended for persons having undergone COVID-19, at least for patients who were hospitalized when being ill (S3 and S4 Tables) [25].

The relatively larger change in health care use in the younger age groups may partly be explained by the fact that they might need a sick note from their GP to claim sick money from the national insurance scheme if their sickness spell exceeds 4 or 7 days. In a sub-analysis we found that removing visits where a sick note was issued, lead to smaller increases in health care use for the younger age groups but hardly changed the results for those typically retired (aged 70 years or older). However, the main results remain unchanged (S2 Table). Note though, that a sick note will be issued in many consultations as a result of the patient's health conditions as revealed by the GP, and thus, excluding all such consultations will surely underestimate the number of patients seeking medical treatment (beyond sickness absence from work). The main takeaway from this analysis is thus limited to observing that the increase in health care use goes beyond the need for GP certification of sickness absence from work.

Furthermore, we had no cut-off for when COVID-19 started or ended, which may explain the increased health care use in the first 1–4 weeks after test, but not in the 2–6 months after test. Considering patient-reported descriptions of COVID-19 feeling like a heavy and long-lasting flu that has different durations for different patients [26], we chose not to set a cut-off for the end of the disease. Rather, we focused on the test date and could shed light on the time frame for the need for health care also when the infection was still ongoing.

## Potential limitations

Some important limitations should be mentioned. First, there was limited test capacity in Norway during the first 3 months of the pandemic [27]. For this reason, we might have missed a large part of the earliest mild COVID-19 cases, and the persons tested in the early spring of 2020 were a selected group more likely to have had severe symptoms and in need of medical treatment. From the summer of 2020, indications for testing were adjusted and test capacity became sufficient to allow and encourage all residents free PCR-tests whenever they had or feared possible symptoms [28]. In addition to using those testing negative as a comparison

group, we could also have matched non-tested residents to those testing positive, with all the well-known advantages and disadvantages of matching procedures. However, because the data and DiD-method allowed us to use a comparison group of persons with actual negative test, we expect test capacities to have affected the groups with COVID-19 and no COVID-19 to a similar extent. Given the very similar pre-test-trends for those with mild and no COVID-19 (see Fig 1), we expect no over- or underestimation of results for the group having mild COVID-19. A second limitation is that we lacked patient-reported data to differentiate between asymptomatic and symptomatic individuals, and thus, we could not identify persons with symptoms that did not lead to visits in health care. However, we believe our analysis can shed light on health care use independent of the nature and severity of complaints.

Finally, our analyses are obviously based on observational data, and what we refer to as impacts of COVID-19 on health care use may of course be related to confounders. While the DiD model accounts well for time invariant individual characteristics, as well as time varying health care utilization that also affects persons with no COVID-19, we cannot be certain that the temporal pattern in utilization of those with no COVID-19 is a reasonable counterfactual for the health care utilization of patients with COVID-19. We do observe, though, that the utilization rate for health care in the months before the test week is almost identical in the group with no and mild COVID-19 (Fig 1), which is what we would have expected if mild COVID-19 were randomly attributed in the test week. We also observe that the health care use of those with mild COVID-19 tend to return to its own pre-test level within our observation window for most of the groups we have studied. Overall, this supports our main finding of only small or no elevated use of health care services for patients with mild COVID-19 a few months after initial infection.

## Conclusion

For the vast majority of patients who are *not* hospitalized with COVID-19, the elevated use of health care services following positive test for SARS-CoV-2 vanishes 2–3 months after testing positive. Women and men at all ages were more or less similarly affected by undergoing a mild disease course (2 months), yet middle-aged women had the most prolonged increased primary care use, which persisted for up to 3 months.

## Supporting information

**S1 Table. Definitions of the cause-specific diagnosis groups applied.**
(DOCX)

**S2 Table. Difference-in-differences (DiD) estimates of impacts of COVID-19 on *primary health care use (without obtaining a sick note)* 12–1 weeks before PCR test for SARS-CoV-2 to 1–4, 5–8, and 9–12 weeks after as well as 13–24 weeks after PCR test.** The DiD estimates captures the change in health care use from 12–1 weeks before PCR test to 1–4, 5–8, and 9–12 weeks after as well as 13–24 weeks after PCR test for *patients with mild COVID-19* compared with the change over the same period for patients with no COVID-19.
(DOCX)

**S3 Table. Difference-in-differences (DiD) estimates of impacts of COVID-19 on health care use from 12–1 weeks before PCR test for SARS-CoV-2 to 1–4, 5–8, and 9–12 weeks after as well as 13–24 weeks after PCR test.** The DiD estimates captures the change in health care use from 12–1 weeks before PCR test to 1–4, 5–8, and 9–12 weeks after as well as 13–24

weeks after PCR test for *women patients with severe* COVID-19 compared with the change over the same period for women patients with no COVID-19.
(DOCX)

**S4 Table. Difference-in-differences (DiD) estimates of impacts of COVID-19 on health care use from 12–1 weeks before PCR test for SARS-CoV-2 to 1–4, 5–8, and 9–12 weeks after as well as 13–24 weeks after PCR test.** The DiD estimates captures the change in health care use from 12–1 weeks before PCR test to 1–4, 5–8, and 9–12 weeks after as well as 13–24 weeks after PCR test for *men patients with severe COVID-19* compared with the change over the same period for men patients with no COVID-19.
(DOCX)

## Acknowledgments

We would like to thank the Norwegian Directorate of Health, in particular Director for Health Registries Olav Isak Sjøflot and his department, for excellent cooperation in establishing the emergency preparedness register. We would also like to thank Gutorm Høgåsen and Anja Elsrud Schou Lindman for their invaluable efforts in the work on the register. We would also like to thank Anja Elsrud Schou Lindman, Thor Indseth, Siri Eldevik Håberg, Hanne Løvdal Gulseth, Kjetil Gundro Brurberg, Atle Fretheim and Karin Maria Nygård for critically evaluating the content of the study. The interpretation and reporting of the data are the sole responsibility of the authors and no endorsement by the register is intended or should be inferred.

## Author Contributions

**Conceptualization:** Kjetil Elias Telle, Karin Magnusson.

**Formal analysis:** Katrine Damgaard Skyrud, Kjersti Helene Hernæs, Karin Magnusson.

**Methodology:** Katrine Damgaard Skyrud, Kjetil Elias Telle, Karin Magnusson.

**Project administration:** Kjetil Elias Telle.

**Supervision:** Kjetil Elias Telle, Karin Magnusson.

**Visualization:** Katrine Damgaard Skyrud.

**Writing – original draft:** Kjetil Elias Telle, Karin Magnusson.

**Writing – review & editing:** Katrine Damgaard Skyrud, Kjersti Helene Hernæs, Kjetil Elias Telle, Karin Magnusson.

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
