## [Decision Letter · Decision Letter 0]

16 Jul 2021

PONE-D-21-20514

Impacts of long COVID on elevated use of primary and specialist health care services: a nation wide register study from Norway

PLOS ONE

Dear Dr. Skyrud,

Thank you for submitting your manuscript to PLOS ONE. After careful consideration, we feel that it has merit but does not fully meet PLOS ONE’s publication criteria as it currently stands. Therefore, we invite you to submit a revised version of the manuscript that addresses the points raised during the review process.

Please address the issues and revise accordingly.

We look forward to receiving your revised manuscript.

Kind regards,

Academic Editor

PLOS ONE

Journal Requirements:

[The study was funded by the Norwegian Institute of Public Health. No external funding was received.]

 [The authors received no specific funding for this work]

3. Please upload a copy of Figure 3, to which you refer in your text on page 21. If the figure is no longer to be included as part of the submission please remove all reference to it within the text.

Reviewers' comments:

Reviewer's Responses to Questions

**Comments to the Author**

1. Is the manuscript technically sound, and do the data support the conclusions?

Reviewer #1: Yes

Reviewer #2: Yes

2. Has the statistical analysis been performed appropriately and rigorously? 

Reviewer #1: Yes

Reviewer #2: I Don't Know

3. Have the authors made all data underlying the findings in their manuscript fully available?

Reviewer #1: Yes

Reviewer #2: Yes

4. Is the manuscript presented in an intelligible fashion and written in standard English?

Reviewer #1: Yes

Reviewer #2: Yes

5. Review Comments to the Author

Reviewer #1: This is a nationwide register study looking at the impact of mild COVID-19 on the use of health care service. The study is based on high quality data from national Norwegian registers.

The title is somehow misleading as it is studying mild COVID (defined as not hospitalized) and not long COVID.

In the abstract it is stated that all persons tested for SARS-CoV-2 is included. However, only adults > 20 years with a PCR-test is included. The use of antigen-test in Norway is not discussed.

The studyperiod covers the first year of the pandemic. In that period the PCR-testing capacity and the indication for taking at test may have changed. Initial it was primarily because of symptoms, but later it may be a requirement for a negative test to get a "Coronapass". This may affect the use of health services after a test.

Why did the authors not compare use of health care services with the background population without a COVID-19 test, match on age, gender and recidency?

The percentage of persons born abroad is nearly twice as high in the group with a positive test than in the group with a negative test. The authors do not adress this difference. Could cultural differences in the use of health care affect the results? The results confirms results from other European countries showing that som ethnic groups are more affected by the COVID-19 epidemic.

The result of the stydy is very reassuring that the majority of people with mild COVID-19 don´t have longlasting symptoms with need for medical attention.

Reviewer #2: Interesting study analysing a clinical important question.

It is reasuring that the authors find that the increase in usage of primary health care after mild COVID-19 is temporary and that there is no increase in usage of specialists.

Some of the findings are counter-intuitive i.e. the finding that the increased usage of primary care is lower among the eldest patients relative to those aged 45-69. This findings needs to be discussed in more detail. Could it be due to the patients of "working-age" having need for i.e. documentation for sick leave, while this is not the case for those already retired?

It is stated that half of the concultations were digital (page 21), could this be explained in more detail, is it phone consultation, video- or e-mail consultations, and did the frequencies of digital consultation vary according to age group and timing from infection?

In the result section it is stated that n equals 1,401,922. Is this the first test ever for the individuals? What happens to people testing first negative and later positive or vice versa?

Page 9, paragraph 3, it is stated that 0.74 % of those with mild COVID-19 died within 3 months vs 0.30 % of those testing negative. Was this difference significant?

In Tabl1 the percentages of individuals born abroad is listed. How does this relate to national numbers? Is this information used? What is the impact of been born outside of Norway? If these data are listed, they need to be discussed.

On page 5 sentence 94 suggest including 2020 after March 1st

6. PLOS authors have the option to publish the peer review history of their article (what does this mean?). If published, this will include your full peer review and any attached files.

Reviewer #1: No

Reviewer #2: No

---

## [Author Response · Author response to Decision Letter 0]

1 Sep 2021

We would like to thank the expert reviewers for valuable input, which has helped to improve the quality of our manuscript. Please find below a point-to-point response to your comments and a list of the changes we made in the revised manuscript. 

Comments of reviewer 1 Our response Action

This is a nationwide register study looking at the impact of mild COVID-19 on the use of health care service. The study is based on high quality data from national Norwegian registers.

The title is somehow misleading as it is studying mild COVID (defined as not hospitalized) and not long COVID.

 We agree. We have revised our title into:

“Impacts of mild COVID-19 on elevated use of primary and specialist health care services: a nationwide register study from Norway”

In the abstract it is stated that all persons tested for SARS-CoV-2 is included. However, only adults > 20 years with a PCR-test is included. Thank you for this clarifying comment. We agree this should be specified. We have made the following changes to the Methods (in Abstract): “In all adults (≥20 years) tested for SARS-CoV-2 in Norway March 1st 2020 to February 1st 2021 (N=1 401 922). “

Under Methods in the manuscript, we added under 1) Population: 

“Our population included every adult (≥20 years) resident of Norway on January 1st 2020 ……”

And 2) COVID-19:

We studied all adults (≥20 years) with a PCR-tests for SARS-CoV-2 in Norway, ….”

The use of antigen-test in Norway is not discussed. Yes, we only have data for PCR-tests, and we agree that the use of antigen-tests in Norway could have been better discussed. New text added under Methods: Design and data sources on pp. 4-5:

“Anti-gen tests were rarely used before February 2021, which is when our study period ends, and all positive anti-gen tests were followed up with a PCR test and thus included in our analyses.”

The studyperiod covers the first year of the pandemic. In that period the PCR-testing capacity and the indication for taking at test may have changed. Initial it was primarily because of symptoms, but later it may be a requirement for a negative test to get a "Coronapass". This may affect the use of health services after a test.

 Yes, test criteria changed, especially in the first months of the pandemic. From March to August 2020, testing was restricted to the elderly, persons at risk (with underlying medical conditions) and health personnel. From August 2020 test capacity was much higher, and everyone with symptoms or who suspected they had COVID-19 could be tested. The “coronapass” was introduced in Norway in May/June 2021, i.e. outside our study period and should thus have no impact on our findings. We have now added more information about potential impact of differing test criteria, see Potential limitations in the Discussion section, p. 23, which now reads:

“Some important limitations should be mentioned. First, there was limited test capacity in Norway during the first 3 months of the pandemic [27]. For this reason, we might have missed a large part of the earliest mild COVID-19 cases, and the persons tested in the early spring of 2020 were a selected group more likely to have had severe symptoms and in need of medical treatment. From the summer of 2020, indications for testing were adjusted and test capacity became sufficient to allow and encourage all residents free PCR-tests whenever they had or feared possible symptoms [28].” 

Why did the authors not compare use of health care services with the background population without a COVID-19 test, match on age, gender and recidency? Thank you for addressing this important question about the need for always including adequate comparison groups. In this study we chose the persons testing negative as our comparison group. An alternative comparison group would be persons without a COVID-19 test. Both alternatives have strengths and weaknesses. For untested persons we do not have a test date to define a reference point for health care use (to enable comparing post-test use with pre-test use). For the negative persons we do have this test date which makes it straightforward and easy to understand strengths and weaknesses of our approach. We have tried to be transparent on these strengths and weaknesses, see discussion in ??Limitations p ?? If we used non-tested persons in comparison group, we would assign them such a test data with some matching procedure, and there are several adequate ways of doing it though, as with matching procedures in general, there are always pros and cons which are not always very transparent. 

It would of course be possible to do both, i.e. use two reference groups and check that main results are the same. Given the clear temporal pattern in health care use for those testing positive, the temporal pattern in a comparison group would need to be quite peculiar to alter the main results. On this background, and given the deadline for the resubmission, we have not had time to undertake such analyses now. If the reviewer or the editor believe that adding such a comparison group is crucial, we would of course be willing to do so. In trying to further improve transparency in this regard, we have added the following to the discussion section, p. 23: 

“Some important limitations should be mentioned. First, there was limited test capacity in Norway during the first 3 months of the pandemic [27]. For this reason, we might have missed a large part of the earliest mild COVID-19 cases, and the persons tested in the early spring of 2020 were a selected group more likely to have had severe symptoms and in need of medical treatment. From the summer of 2020, indications for testing were adjusted and test capacity became sufficient to allow and encourage all residents free PCR-tests whenever they had or feared possible symptoms [28]. In addition to using those testing negative as a comparison group, we could also have matched non-tested residents to those testing positive, with all the well-known advantages and disadvantages of matching procedures. However, because the data and DiD-method allowed us to use a comparison group of persons with actual negative test, we expect test capacities to have affected the groups with COVID-19 and no COVID-19 to a similar extent. Given the very similar pre-test-trends for those with mild and no COVID-19 (see Figure 1), we expect no over- or underestimation of results for the group having mild COVID-19.”

The percentage of persons born abroad is nearly twice as high in the group with a positive test than in the group with a negative test. The authors do not adress this difference. Could cultural differences in the use of health care affect the results? The results confirms results from other European countries showing that som ethnic groups are more affected by the COVID-19 epidemic. Yes, the percentage of individuals born abroad is about 19% in total in Norway, compared to about 35% for those with mild COVID-19 (Table 1). Several Norwegian and European studies have shown higher infection rate for persons born abroad. We do not have any data to straightforwardly capture cultural differences, but we agree that this is very interesting and important issues. We have added the following paragraph to the discussion section, p. 20: 

“Our study revealed a higher infection rate for persons born outside Norway (Table 1), and our findings are supported by other European studies [20-21]. Labberton et al. show that COVID-19 lead to higher rates of infection and hospitalization in Norway’s immigrant population [22]. Thus, although country of origin remains a relevant factor when studying healthcare use, the subject deserves a more thorough investigation, which is beyond the scope of this article.

The result of the stydy is very reassuring that the majority of people with mild COVID-19 don´t have longlasting symptoms with need for medical attention. Yes, we agree. 

Reviewer 2 Our response Action

Interesting study analysing a clinical important question. Thank you. 

It is reasuring that the authors find that the increase in usage of primary health care after mild COVID-19 is temporary and that there is no increase in usage of specialists. Yes, we agree. 

Some of the findings are counter-intuitive i.e. the finding that the increased usage of primary care is lower among the eldest patients relative to those aged 45-69. This findings needs to be discussed in more detail. Could it be due to the patients of "working-age" having need for i.e. documentation for sick leave, while this is not the case for those already retired? Thanks for pointing out this finding and we agree that this is worth discussing in more detail. We believe there are especially two explanations for this finding. One may be that, in general, the younger age groups used less health care before COVID-19, then the relative increase in use of primary health care after COVID-19 would be higher for the younger than the older age groups. In addition, as the reviewer nicely points out, it may be due to patients younger than 70 years of age will be in need of doctor-certified sick leave (sick note). This could in fact be explored using our data. 

 We included a sub-analysis on primary care for persons with mild COVID-19 where we excluded consultations resulting in doctor-certified sick leave (sick notes). 

This is now specified in Statistical analysis (paragraph 5, pages 8-9):

“Some patients may contact the primary care physician to certify sickness absenteeism from work, and to explore the robustness of our results, we also undertook an analysis where we excluded consultations that resulted in the physician providing a sick note. “

And in the Discussion under Interpretation (paragraph 3, p 22):

“The relatively larger change in health care use in the younger age groups may partly be explained by the fact that they might need a sick note from their GP to claim sick money from the national insurance scheme if their sickness spell exceeds 4 or 7 days. In a sub-analysis we found that removing visits where a sick note was issued, lead to smaller increases in health care use for the younger age groups but hardly changed the results for those typically retired (aged 70 years or older). However, the main results remain unchanged. Note though, that a sick note will be issued in many consultations as a result of the patient’s health conditions as revealed by the GP, and thus, excluding all such consultations will surely underestimate the number of patients seeking medical treatment (beyond sickness absence from work). The main takeaway from this analysis is thus limited to observing that the increase in health care use goes beyond the need for GP certification of sickness absence from work.” 

It is stated that half of the concultations were digital (page 21), could this be explained in more detail, is it phone consultation, video- or e-mail consultations, and did the frequencies of digital consultation vary according to age group and timing from infection?

 Yes, the 50% refers to the percentage of consultations in the test week which were digital. In total 30% of the consultations were digital. As we would expect, we’ve checked and the frequency of digital consultations are the highest in the youngest age group (about 30% compared to 18% in the oldest age group). We were not able to distinguish between phone, video-or e-mail consultations. In addition, during the pandemic the payment received by the doctor was increased for digital consultations. Based on the comment from the reviewer we revised the following in the Discussion under Interpretation, p. 21:

“Of further importance to the interpretation of our findings are the high rate of primary care use during the test week. These peaks are expected, as we also include digital general practitioner (GP) consultations (phone, video-or e-mail consultations). About 50% of all consultations in the test week were digital, compared to 30% thereafter, and were more frequent in the younger age groups. For example, digital GP consultations are likely to increase when COVID-19 is suspected and patients are quarantined or isolated. In addition, in the spring of 2020 the doctors’ payments for conducting digital consultations rose to the same level as traditional consultations, and it was allowed to issue sick notes in digital consultation [23]. We were not able to distinguish between phone, video-or e-mail consultations.”

In the result section it is stated that n equals 1,401,922. Is this the first test ever for the individuals? What happens to people testing first negative and later positive or vice versa? Thank you for this clarifying comment. We agree this could have been better described. We have rephrased the sentence under point 2 on page 5 as the following: 

“The positive test may or may not be preceded by one or more negative tests. For the very few individuals with several positive tests, we used the first.”

Page 9, paragraph 3, it is stated that 0.74 % of those with mild COVID-19 died within 3 months vs 0.30 % of those testing negative. Was this difference significant? Thanks for the comment. We provide the confidence interval (paragraph 3, p. 9) and they are not overlapping, suggesting that the difference is significant.

In Tabl1 the percentages of individuals born abroad is listed. How does this relate to national numbers? Is this information used? What is the impact of been born outside of Norway? If these data are listed, they need to be discussed. Yes, the percentage of individuals born abroad is about 19% in total in Norway, compared to about 35% for those with mild COVID-19 (Table 1). Several Norwegian and European studies have shown higher infection rate for persons born abroad. Although country background remains a relevant factor when studying healthcare use, the subject deserves a more thorough investigation, which is beyond the scope of this article. Please see comment from R1. 

 We have added the following paragraph to the discussion section, p. 20: 

“Our study revealed a higher infection rate for persons born outside Norway (Table 1), and our findings are supported by other European studies [20-21]. Labberton et al. show that COVID-19 lead to higher rates of infection and hospitalization in Norway’s immigrant population [22]. Thus, although country of origin remains a relevant factor when studying healthcare use, the subject deserves a more thorough investigation, which is beyond the scope of this article.

On page 5 sentence 94 suggest including 2020 after March 1st We agree. Corrected.

---

## [Decision Letter · Decision Letter 1]

14 Sep 2021

Impacts of mild COVID-19 on elevated use of primary and specialist health care services: a nationwide register study from Norway

PONE-D-21-20514R1

Dear Dr. Skyrud,

We’re pleased to inform you that your manuscript has been judged scientifically suitable for publication and will be formally accepted for publication once it meets all outstanding technical requirements.

Kind regards,

Academic Editor

PLOS ONE

Additional Editor Comments (optional):

Reviewers' comments:

Reviewer's Responses to Questions

**Comments to the Author**

1. If the authors have adequately addressed your comments raised in a previous round of review and you feel that this manuscript is now acceptable for publication, you may indicate that here to bypass the “Comments to the Author” section, enter your conflict of interest statement in the “Confidential to Editor” section, and submit your "Accept" recommendation.

Reviewer #1: All comments have been addressed

Reviewer #2: All comments have been addressed

2. Is the manuscript technically sound, and do the data support the conclusions?

Reviewer #1: Yes

Reviewer #2: Yes

3. Has the statistical analysis been performed appropriately and rigorously? 

Reviewer #1: Yes

Reviewer #2: Yes

4. Have the authors made all data underlying the findings in their manuscript fully available?

Reviewer #1: Yes

Reviewer #2: Yes

5. Is the manuscript presented in an intelligible fashion and written in standard English?

Reviewer #1: Yes

Reviewer #2: Yes

6. Review Comments to the Author

Reviewer #1: (No Response)

Reviewer #2: Thank you for your thorough response to the issues raised.

I find that the current version is suitable for publication.

7. PLOS authors have the option to publish the peer review history of their article (what does this mean?). If published, this will include your full peer review and any attached files.

Reviewer #1: **Yes: **Carsten Schade Larsen

Reviewer #2: No

---

## [Editor Report · Acceptance letter]

30 Sep 2021

PONE-D-21-20514R1 

Impacts of mild COVID-19 on elevated use of primary and specialist health care services: a nationwide register study from Norway 

Dear Dr. Skyrud:

I'm pleased to inform you that your manuscript has been deemed suitable for publication in PLOS ONE. Congratulations! Your manuscript is now with our production department. 

Kind regards, 

on behalf of

Dr. Robert Jeenchen Chen 

Academic Editor

PLOS ONE